# High-Efficiency Spin-Related Vortex Metalenses

**DOI:** 10.3390/nano11061485

**Published:** 2021-06-03

**Authors:** Wei Wang, Ruikang Zhao, Shilong Chang, Jing Li, Yan Shi, Xiangmin Liu, Jinghua Sun, Qianlong Kang, Kai Guo, Zhongyi Guo

**Affiliations:** 1Department of Mathematics and Physics, Shijiazhuang Tiedao University, Shijiazhuang 050043, China; wangw@stdu.edu.cn (W.W.); zrk111@live.com (R.Z.); 17734562693@163.com (S.C.); jingli@stdu.edu.cn (J.L.); mrshiyan@163.com (Y.S.); liuxy424@163.com (X.L.); 2School of Electrical Engineering and Intelligentization, Dongguan University of Technology, Dongguan 523808, China; sunjh@dgut.edu.cn; 3School of Computer and Information, Hefei University of Technology, Hefei 230009, China; 2018110992@mail.hfut.edu.cn (Q.K.); kai.guo@hfut.edu.cn (K.G.)

**Keywords:** metasurfaces, vortex beam, vortex metalenses, topological charge

## Abstract

In this paper, one spin-selected vortex metalens composed of silicon nanobricks is designed and numerically investigated at the mid-infrared band, which can produce vortex beams with different topological charges and achieve different spin lights simultaneously. Another type of spin-independent vortex metalens is also designed, which can focus the vortex beams with the same topological charge at the same position for different spin lights, respectively. Both of the two vortex metalenses can achieve high-efficiency focusing for different spin lights. In addition, the spin-to-orbital angular momentum conversion through the vortex metalens is also discussed in detail. Our work facilitates the establishment of high-efficiency spin-related integrated devices, which is significant for the development of vortex optics and spin optics.

## 1. Introduction

Metasurfaces are made up of subwavelength scatterers and possess great potential for developing ultrathin optics [1,2,3,4,5,6,7,8]. In addition, metasurfaces are two-dimensional planar nanostructures and are more suitable for integration devices. Different from conventional optical devices, metasurfaces can manipulate the amplitude, wavefront and polarization state of light with extreme freedom. By altering the geometry size and orientation of unit cells, metasurfaces can shape the wavefront of the scattered light to the desired form. Metasurfaces demonstrate the strong wavefront control capabilities and can be used to develop various optical devices, including beam deflectors [9,10], metalenses [11,12,13,14,15,16], holograms [17,18,19,20], vortex generators [21,22,23,24,25], and absorbers [26,27]. Additionally, metasurfaces have received considerable interest in the field of spin photonics. Spin-selective optical manipulation plays an important role in spin photonics, quantum optics and integrated optics. Recently, spin-selective transmission [28,29], beam deflection [30,31], focusing [32,33] and holograms [34] have been successively implemented with metasurfaces.

Optical vortices have been a burgeoning research area in the past few decades since their discovery in 1992 [35]. Different from plane waves, vortex beams possess helical phase fronts characterized by exp(ilφ) and orbital angular momentum (OAM) of lℏ, where l is the topological charge, φ is the azimuthal angle and ℏ is the reduced Planck’s constant. Compared to spin angular momentum (SAM) with two values of ±ℏ. OAM can take on an arbitrary value. Therefore, vortex beams are extensively used in optical communication [36,37], quantum systems [38] and particle trapping [39]. Benefiting from the ultrathin and miniaturized nature of metasurfaces, a variety of vortex beam generators based on metasurfaces have been designed and studied [40,41,42,43]. Nevertheless, most vortex beam generators only produce one specific topological charge. Moreover, most vortex beam generators always generate a propagating vortex beam rather than converged vortex beams, which is also very important in many optical applications, such as manipulate microparticles. The vortex metalens can produce the converged vortex beam with a focal plane, which can be designed by integrating the functions of vortex generators and lenses into a single metasurface. Generally speaking, vortex metalens based on the Pancharatnam−Berry (PB) phase is a bipolar metalens, which only achieve one kind of spin light [44,45,46,47]. Through multiplexing two opposite polarity vortex metalenses based on the PB phase, the focusing vortex beam with arbitrary topological charge can be produced for two opposite spin lights [44,48]. However, under left circularly polarized (LCP) or right circularly polarized (RCP) light incidence, one part of the metasurface units work and produce the convergent vortex beam; other metasurface units do not work or play the opposite role. Therefore, the multiplexed vortex metalens has an inevitably high signal-to-noise ratio and low efficiency, as the theory limit of highest efficiency is 50%. This limits the practical application of spin-related vortex metalenses. In addition, as far as we know, there have been no reports on high-efficiency spin-related vortex metalenses. Meanwhile, the spin-to-orbital angular momentum conversion through the vortex metalens has rarely been discussed before. On the other hand, compared to mid-infrared wavelengths, dielectric metasurfaces have been extensively studied in visible and near-infrared wavelengths. However, the mid-infrared band also has a lot of potential value, such as space communication, molecule exploration and imaging.

In this paper, we propose two high-efficiency broadband spin-related vortex metalenses based on silicon nanobricks with high transmission efficiency, the design of which is based on both propagation and geometric phases, whereby each unit plays a positive role under LCP or RCP light incidence. Our designed spin-selected vortex metalens can focus the vortex beams with different spin states to the same position at the wavelength of 4500 nm, respectively, and the vortex beams possess different topological charges. We also designed one spin-independent vortex metalens, in which the vortex beams can possess the same topological charge and the same focusing plane for different spin lights. The broadband characteristic of a vortex metalens is also studied, and the work bandwidth is measured at about 1500 nm (in the range of 4000 nm–5500 nm). At the same time, the spin-to-orbital angular momentum conversion through the vortex metalens is also discussed. These spin-related optical vortex metalenses are significant for communication systems and spin-controlled photonics.

## 2. Theoretical Analysis and Design Methodologies

To design a high-efficiency spin-related optical vortex metalens without functional crosstalk, the metasurfaces are composed of subwavelength Si nanobricks on a CaF_2_ substrate with a refractive index of 1.4. Si is a high-refractive index dielectric material in the mid-infrared range, which possesses a phase modulation ranging from 0 to 2π. The complex permittivity of Si is extracted from data from Pierce [49]. Figure 1a shows a Si nanobrick with three independent geometry parameters of (L, W, *θ*) and a constant height H. The rectangular Si nanobricks possess different effective refractive indices along the two axes, which can produce independent phases for two orthogonal linear polarizations. The transmittance of the Si nanobrick can be described using the Jones matrix as [50]:(1)$$T=R(-\theta )\left(\begin{array}{l} \begin{array}{cc} t_{xx}e^{i\varphi_{xx}} & 0 \end{array}\\ \begin{array}{cc} 0 & t_{yy}e^{i\varphi_{yy}} \end{array} \end{array}\right)R(\theta )$$
where *R*(*θ*) is the rotation matrix, $t_{xx}$ and $\varphi_{xx}$ are the X linearly polarized (XLP) transmission coefficient and propagation phase, and $t_{yy}$ and $\varphi_{yy}$ are the Y linearly polarized (YLP) transmission coefficient and propagation phase, respectively. The relation between the electric fields of the input and output light is $E_{out}=TE_{in}$. Under the incidence of circularly polarized (CP) light, the nanobrick should be a half-wave plate for achieving the high polarization conversion efficiency.

The cross-polarized transmitted phases can be deduced as $\Phi_{RL}=\varphi_{xx}+2\theta$ and $\Phi_{LR}=\varphi_{xx}-2\theta$, respectively. Here, the nanobrick can work in LCP and RCP incidences, simultaneously. The selection of nanobricks can be expressed as
(2)$$\varphi_{xx}=\frac{1}{2}\left[\left(\Phi_{LR}-2n_{1}\pi \right)+\left(\Phi_{RL}-2n_{2}\pi \right)\right]$$
(3)$$\theta =\frac{1}{4}\left[\left(\Phi_{RL}-2n_{2}\pi \right)-\left(\Phi_{LR}-2n_{1}\pi \right)\right]$$
where $n_{1}$ and $n_{2}$ are integers. Therefore, a single nanobrick can completely control polarization and phase by choosing the suitable structure parameters. The amplitudes and phase shifts of the irrational nanobrick under the XLP and YLP incidences have been simulated by the finite difference time domain (FDTD) method at the wavelength of 4500 nm, as shown in Figure 1b–e. The period of the unit was set as 2100 nm, and the height of the nanobrick was 4000 nm to ensure all possible propagation phase combinations from 0 to 2π. The total phases of the spin-related optical vortex metalens, which can focus both the LCP and RCP light into vortex beams, can be described as follows:(4)$$\{\begin{array}{c} \Phi_{lcp}=\frac{2\pi }{\lambda }\left(\sqrt{x^{2}+y^{2}+f_{l}^{2}}-f_{l}\right)+l_{l}\cdot a\tan(y/x)\\ \Phi_{rcp}=\frac{2\pi }{\lambda }\left(\sqrt{x^{2}+y^{2}+f_{r}^{2}}-f_{r}\right)+l_{r}\cdot a\tan(y/x) \end{array}$$
where λ is the incident wavelength, $f_{l}$ and $f_{r}$ are the focal length corresponding to LCP and RCP incidences, respectively, and $l_{l}$ and $l_{r}$ are the topological charge corresponding to LCP and RCP incidences, respectively. According to Equations (2)–(4), the different phase responses of the optics vortex metalens for LCP and RCP light can be manipulated independently and simultaneously by changing the geometrical parameters of the nanobricks. Furthermore, the optics vortex metalens possesses spin-selectivity and high efficiency.

## 3. Results and Discussion

The spin-selected vortex metalenses can simultaneously focus both LCP and RCP light into converged cross-polarized vortex beams with different topological charges. The designed parameters are λ = 4500 nm, $f_{l}$ = $f_{r}$ = 12 um, $l_{l}$ = 1, $l_{r}$ = 2, respectively. Firstly, according to Equation (4), the phases ($\;\Phi_{lcp},\Phi_{rcp}$) at any position of the vortex metalens under LCP and RCP incidences can be obtained. Secondly, the propagation phase $\varphi_{xx}$ and rotation angle θ can be obtained, according to Equations (2) and (3). Lastly, according to Figure 1b–e, the specific length and width (L,W) of the nanobrick can be easily selected. The schematic diagram and top view of the designed spin-selected vortex metalens are shown in Figure 2a,b, respectively.

Under the LCP incidence, Figure 3a shows the donut-shaped electric field intensity distribution at the focal plane. Figure 3b shows the corresponding intensity distribution in the x-z plane with two symmetrically distributed focal points, which illustrates the vortex beam as being approximately focused in the preset position. Figure 3c shows the phase distribution at the focal plane, which means that the topological charge is l = 1 for the generated vortex beam. Under the RCP incidence, Figure 3d,e show the electric field intensity distribution at the focal plane and x-z plane, respectively, and Figure 3f shows the phase distribution at the focal plane, which demonstrates that the converged vortex beam with l = 2 is generated at a nearly preset position. Therefore, the converged vortices with different topological charges can be formed through our designed vortex metalens for the LCP and RCP incidences, respectively. Different from multiplexed optical vortex metalens, our designed vortex metalens can achieve a high efficiency of 70.3% and 68.4% under LCP and RCP incidence, respectively.

Light can transport angular momentum via two components, namely SAM and OAM; the formation of the vortex beam is generally related to the interaction of SAM and OAM. Generally, it is believed that the vortex generator based on the PB phase can produce spin-dependent vortex beams, and that the interaction of SAM and OAM of light is symmetric for two incidences with opposite spins. However, for our designed spin-selected optical vortex metalens, the interaction of SAM and OAM of light is asymmetric for two incidences with opposite spins. Under the LCP light incidence, the SAM of incident light is ℏ, and incident OAM is 0, while the SAM of output light is −ℏ, and output OAM is ℏ. Therefore, the optical vortex metalens obtains the angular momentum of ℏ based on the angular momentum conservation. Under the RCP light incidence, the SAM of incident light is −ℏ, and incident OAM is 0, while the SAM of output light is ℏ, and output OAM is 2ℏ. Therefore, the optical vortex metalens obtains the angular momentum of −4ℏ. It should be noted that if the designed vortex metalens is freestanding and tiny enough, it will rotate under the incidences of both LCP and RCP light. In any case, the asymmetrical interaction between the SAM and OAM will promote the development of multi-function devices.

The spin-independent vortex metalens was also designed and investigated, which demonstrated that the cross-polarized vortex beams with the same topological charges can be focused on the same positions for both LCP and RCP incidences. The designed parameters are set as λ = 4500 nm, $f_{l}$ = $f_{r}$ = 9 um, $l_{l}$ = $l_{r}$ = 1, respectively. The schematic diagram and top view of the designed spin-independent vortex metalens are shown in Figure 4a,b, respectively. In addition, the design concept of the arbitrarily located nanobrick is shown in detail in Figure 4b.

Under LCP and RCP light incidence, Figure 5a,d show the donut-shaped electric field intensity distributions at the focal plane. Figure 5b,e show the corresponding intensity distributions in the x-z plane. Figure 5c,f show the phase distributions at the focal plane. As expected, the vortex beams with the topological charges l = 1 are focused very well at the almost preset position, which is 8.5 μm behind the metalens. Under the LCP light incidence, incident SAM and OAM are ℏ and 0, respectively. While output SAM is −ℏ, output OAM is ℏ. Under the RCP light incidence, the incident SAM is −ℏ, and incident OAM is 0, while output SAM is ℏ, output OAM is ℏ. The optical vortex metalens obtains the angular momentum of ℏ and −3ℏ under the LCP and RCP light incidence, respectively, and the focusing efficiency reaches up to 58.2% and 66% for LCP and RCP incidences, respectively. Under the XLP light incidence, as shown in Figure 5g–i, there is the same focusing vortex beam as CP incidence. Therefore, the optics vortex metalens, which is designed based on the propagation phase and geometric phase, has high-efficiency polarization-independent properties for CP and LP incident lights.

We also studied the multispectral characteristics of our designed spin-independent vortex metalens. Under the LCP incidence with the wavelength of 5500 nm, the focal lengths reduced quickly with the increase in wavelength, as shown in Figure 6b. Figure 6a,c show the electric field intensity and phase distribution at the focal plane, respectively, which demonstrate that the vortex beam with the topological charge of l = 1 is produced. Under the RCP incidence with the wavelength of 4000 nm, Figure 6d,e show the electric field intensity at the focal plane and x-z plane, and Figure 6f shows the phase distribution at the focal plane, which means the converged vortex beam with l = 1 is produced at higher position. The shift of the focal length originates from the chromatic aberration of the lenses. Therefore, the vortex metalens exhibits excellent broadband characteristics in the mid-infrared band of 4000 nm–5500 nm.

## 4. Conclusions

In summary, we have proposed an approach to design the broadband spin-related vortex metalens that can generate a focusing vortex beam for different spin lights. We have designed one spin-selected vortex metalens, which can focus the LCP and RCP vortex beams with different topological charges to the same position, respectively. We have also designed one spin-independent vortex metalens, which can focus the LCP, RCP and XLP vortex beams with the same topological charge to the same position, respectively. Different from the multiplexing of vortex metalens with a complicated design and high crosstalk, our designed vortex metalenses possess high efficiency and good quality for both LCP and RCP incident light. The interactions between SAM and OAM have also been investigated and discussed in detail. Our proposed approach may not only open a new door for future applications in spin optics and vortex optics, but may also have a profound impact on nanoparticle manipulation and quantum-information processing.

## Figures and Tables

**Figure 1 nanomaterials-11-01485-f001:**
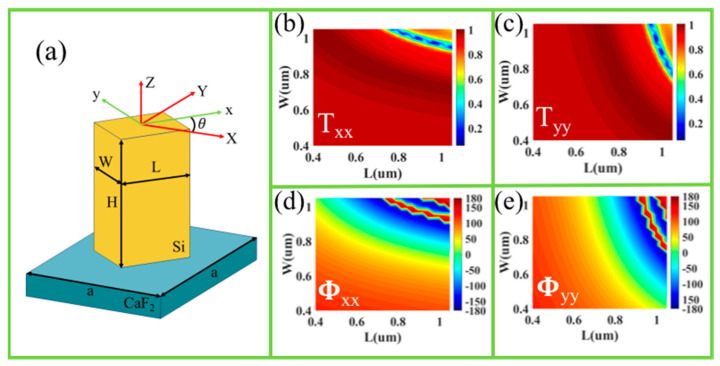
(**a**) Schematic of the Si nanobrick with the rotation angle *θ*. The transmittances (**b**,**c**) and the propagation phases (**d**,**e**) under XLP (**b**,**d**) and YLP (**c**,**e**) incidences of the irrotational Si nanobricks as a function of L and W.

**Figure 2 nanomaterials-11-01485-f002:**
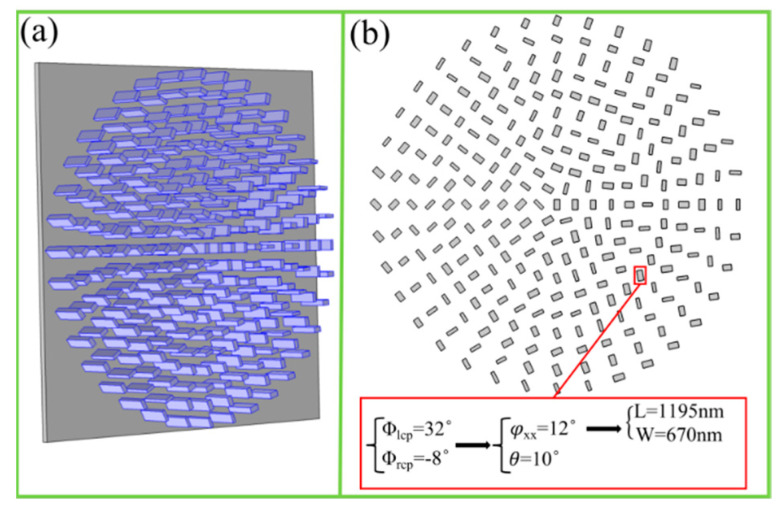
(**a**) Schematic of the spin-selected vortex metalens. (**b**) Top view of spin-selected vortex metalens, and the designing concept and specific parameters’ selection of a unit cell.

**Figure 3 nanomaterials-11-01485-f003:**
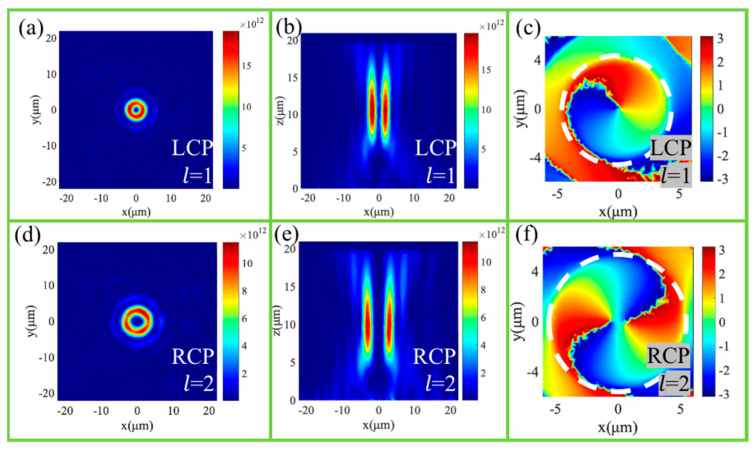
The intensity distributions of the vortex beam at the focusing plane under the LCP (**a**) and RCP (**d**) light illuminations. The intensity distributions of the converged vortex beam at the x-z plane under the LCP (**b**) and RCP (**e**) light illuminations. The phase distributions of the vortex beam at the focusing plane under the LCP (**c**) and RCP (**f**) light illuminations.

**Figure 4 nanomaterials-11-01485-f004:**
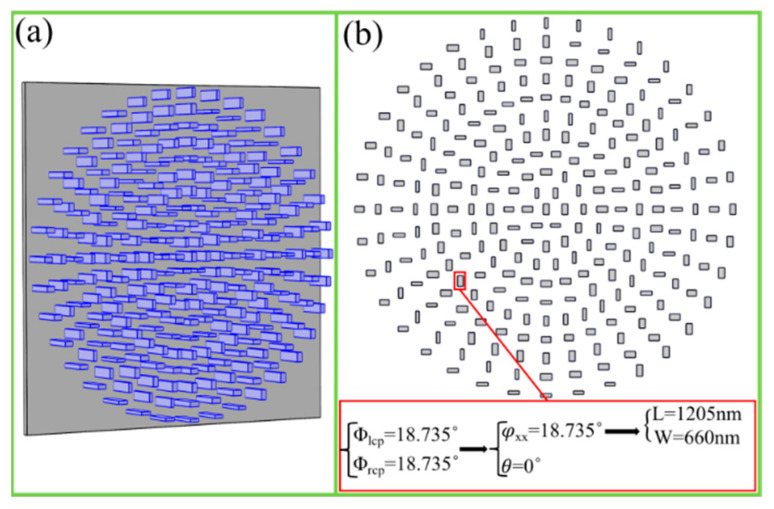
(**a**) Schematic of the spin-independent vortex metalens. (**b**) Top view of spin-independent vortex metalens, and the designing concept and specific parameters’ selection of a unit cell.

**Figure 5 nanomaterials-11-01485-f005:**
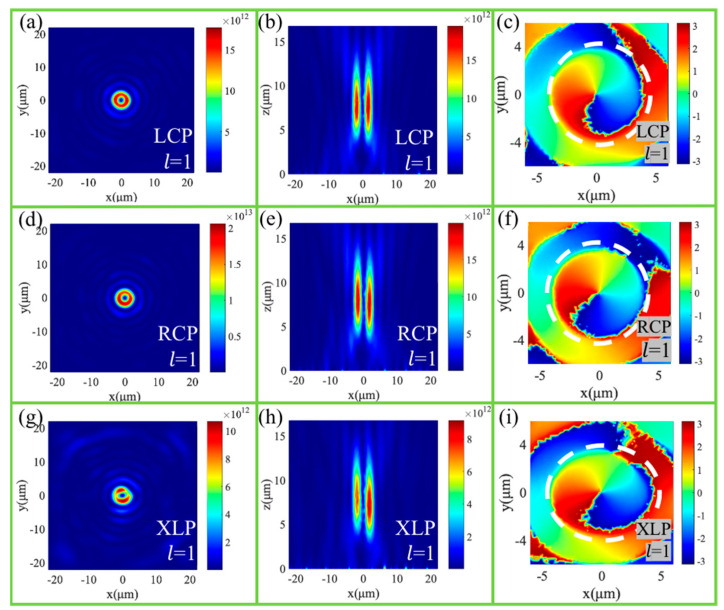
The intensity distributions of the vortex beam at the focusing plane under the LCP (**a**), RCP (**d**) and XLP (**g**) light illuminations. The intensity distributions of the converged vortex beam at the x-z plane under the LCP (**b**), RCP (**e**) and XLP (**h**) light illuminations. The phase distributions of the vortex beam at the focusing plane under the LCP (**c**), RCP (**f**) and XLP (**i**) light illuminations.

**Figure 6 nanomaterials-11-01485-f006:**
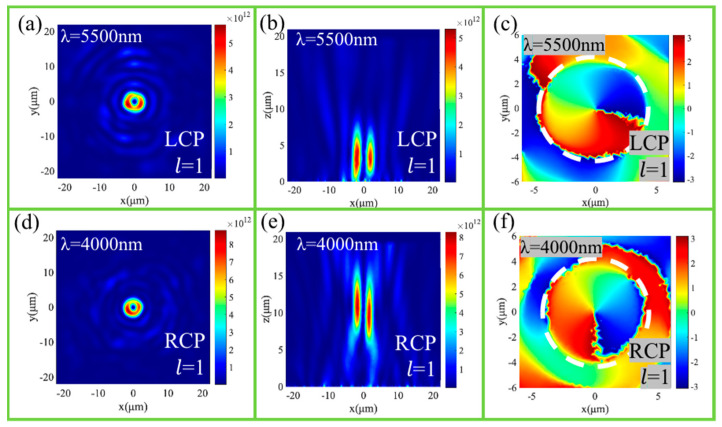
The intensity distributions of the vortex beam at the focusing plane under the LCP (5500 nm) (**a**) and RCP (4000 nm) (**d**) light illuminations. The intensity distributions of the converged vortex beam at the x-z plane under the LCP (5500 nm) (**b**) and RCP (4000 nm) (**e**) light illuminations. The phase distributions of the vortex beam at the focusing plane under the LCP (5500 nm) (**c**) and RCP (4000 nm) (**f**) light illuminations.

## Data Availability

Data available in a publicly accessible repository.

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
