# Peer review of "High-Efficiency Spin-Related Vortex Metalenses"

_nanomaterials, 2021, doi:10.3390/nano11061485_

Round 1

Reviewer 1 Report

The work presents one spin-selected vortex metalens composed of silicon nanobricks. They numerically demonstrate the possibility to produce the vortex beams with different topological charges.

Spin-independent vortex metalens have been proposed. They can focus vortex beams with the same topological charge at the same position for different spin lights. The spin-to-orbital angular momentum conversion through the vortex metalens was also discussed. The work offers new avenues for the development of vortex optics.

I find the paper very interesting and novel and suggest to be accepted. My only comment is related to references. Vortex beams could be also generated by shining light through topological defects (TDs) in thin liquid crystal films. Namely, in these materials, diverse TDs could be stabilized and consequently, diverse vortex beam structures could be created, see e.g.

https://link.aps.org/doi/10.1103/PhysRevResearch.2.013176

Reviewer 2 Report

This paper describes the simulation of formation of optical vortex beams in a microstructure consisting of so-called silicon nanobricks (they are actually several micrometer in size) on a transparent substrate. Depending on the size, geometrical positions and orientation of the Si elements, this structure forms a metalens that can focus or convert different spin lights.

This paper just describes another application of a well-established theoretical framework that is known since about 30 years. Numerous similar simulations have already been published (and by far not only by Chinese groups, as the authors try to pretend when looking at their reference list). Therefore, I don’t see the novelty of this paper. Mere correctness is not sufficient.

The whole approach is very engineering-like. There is actually no materials aspect in this paper: some tabulated values of the optical properties are used, but not even mentioned, any wavelength dependence is ignored. A fixed frequency is assumed, general applicability and limitations are not discussed at all.

The whole paper is designed exclusively for the specialist reader.

E.g. the theory section does not contain all necessary references and information so that one can understand what parameters enter the calculations nor which code has been used.

Unless the authors put more effort in putting their approach into a wider scope and explain to a general readership the originality and novelty of their approach, I cannot recommend publication of this paper.

Reviewer 3 Report

The authors present two designs of metalenses relevant for circularly polarized optical vortices/beams with integer topological charges. The results appear to be scientifically correct.

However, the authors need to stress the originality and novelty of their work.

In its present form, the manuscript is just a variation (and a small one!) of their paper in Results in Physics. There is not enough novelty or originality to deserve publication in Nanomaterials, unless the authors can convince the audience otherwise. As such, the authors should perform a major review to put the paper in a broader context and emphasize eventual applications

Round 2

Reviewer 2 Report

The authors have  revised their manuscript, particularly extended the list of references, and added more information about the validity of their approach.  Unfortunately they have neither extended the introduction nor provided a more readable description of their methods.

In particular, I miss an explicit response my comments. The role of a referee is give (constructive) advice to the authors and the editors, meant to improve the quality of a paper, and I consider this dialogue to be essential. 

Reviewer 3 Report

The authors made some improvements in their revised version. It could now be published